# Nurses’ Involvement in the Development and Usability Assessment of an Innovative Peripheral Intravenous Catheterisation Pack: A Mix-Method Study

**DOI:** 10.3390/ijerph191711130

**Published:** 2022-09-05

**Authors:** Paulo Santos-Costa, Mariana Alves, Carolina Sousa, Liliana B. Sousa, Filipe Paiva-Santos, Rafael A. Bernardes, Filipa Ventura, Anabela Salgueiro-Oliveira, Pedro Parreira, Margarida Vieira, João Graveto

**Affiliations:** 1Health Sciences Research Unit: Nursing (UICISA: E), Nursing School of Coimbra (ESEnfC), 3000-232 Coimbra, Portugal; 2Instituto Ciências da Saúde, Universidade Católica Portuguesa, 4169-005 Porto, Portugal; 3Centro de Investigação Interdisciplinar em Saúde, Universidade Católica Portuguesa, 4169-005 Porto, Portugal

**Keywords:** catheterisation, peripheral, medical device, usability testing, nurses

## Abstract

Guaranteeing peripheral venous access is one of the cornerstones of modern healthcare. Recent evidence shows that the lack of adequate clinical devices can result in the provision of substandard care to patients who require peripheral intravenous catheterization (PIVC). To address this challenge, we aimed to develop a PIVC pack for adult patients and assess the usability of this new device. Methods: Following a mix-method design, the PIVC pack development and usability assessment were performed in two phases with the involvement of its potential end-users (nurses). In phase one (concept and semi-functional prototype assessment), focus group rounds were conducted, and a usability assessment questionnaire was applied at each stage. In phase two (pre-clinical usability assessment), a two-arm crossover randomised controlled trial (PIVC pack versus traditional material) was conducted with nurses in a simulated setting. Final interviews were conducted to further explore the PIVC pack applicability in a real-life clinical setting. Results: High average usability scores were identified in each study phase. During the pre-clinical usability assessment, the PIVC pack significantly reduced procedural time (Z = −2.482, *p* = 0.013) and avoided omissions while preparing the required material (Z = −1.977, *p* = 0.048). The participating nurses emphasised the pack’s potential to standardise practices among professionals, improve adherence to infection control recommendations, and enhance stock management. Conclusions: The developed pack appears to be a promising device that can assist healthcare professionals in providing efficient and safe care to patients requiring a PIVC. Future studies in real clinical settings are warranted to test its cost-effectiveness.

## 1. Introduction

Peripheral intravenous catheterisation (PIVC) is the most common invasive technique performed in hospital settings, with recent estimates showing that all patients require at least one catheter for temporary delivery of intravenous fluids, medications, blood derivatives, or contrast agents. Despite its clinical relevance, PIVC-related complications remain unacceptably common worldwide, being listed as one of the major patient safety concerns in the past year [1]. Recent findings from a comprehensive meta-analysis indicate that patients often experience infiltration/extravasation (13.7%), phlebitis (11%), intraluminal occlusion (8%), pain (6.4%), and catheter dislodgement (6%)[2].

Such complications lead to premature catheter removal and additional insertion attempts, damaging patients’ peripheral vascular network, delaying treatment times, and increasing care costs. This poses a significant challenge for nurses worldwide, who are responsible for catheter insertion, maintenance, and surveillance across most countries [3,4,5]. However, linking PIVC-related outcomes to nurses’ interventions and roles is challenging because they are affected not only by the care provided but also by factors related to patients, interpersonal aspects of care, and the setting in which care is provided [6]. While previous studies have identified risk factors associated with the care provided and patients’ clinical profiles [7,8,9], the availability, quality, and safety of the devices used by nurses in catheter insertion and maintenance are still understudied [10].

In Portugal, as in most international settings, nurses often prepare all the required material before PIVC. In a study by Oliveira and collaborators (2019), nurses often forgot essential catheter insertion and maintenance materials, resulting in unstandardised practices between professionals. Other authors have also found that PIVC is carried out using unsterile or poorly decontaminated materials between patients [11,12,13,14], which can increase the risk of infection. This reality is unacceptable since most items required for catheter insertion and maintenance are relatively low cost [15]. 

In the past years, healthcare managers have focused on the cost-effectiveness of pre-prepared procedural packs in various procedures, from blood collection to surgical interventions [16]. Procedural packs have also gained recent support from international experts on vascular access care, who shared a common understanding that these devices can facilitate an efficient catheter insertion and encourage compliance with best practices [17]. Few studies have addressed the outcomes of implementing a PIVC pack (PIVC-P) in clinical settings [18,19]. Previous findings support that PIVC-P enhances procedure quality and safety, significantly reducing complications such as phlebitis and infiltration [18]. 

However, PIVC-P can present different configurations and designs [17], influencing healthcare professionals’ performance during catheter insertion. In a study conducted in three different hospitals, Franklin and collaborators [20] found that the existing PIVC-P integrated different components, and healthcare professionals are not trained on how to use the packs. This reality restates the importance of usability testing during the development of devices used in vascular access care. As usability testing is an important aspect of patient safety, it is critical that nurses are involved and empowered in designing highly usable devices and making it simpler to deliver safe and high-quality care [21]. 

Thus, we propose to develop a PIVC-P that follows current international standards of care in vascular access in adult patients and can be implemented in clinical settings across Portugal. We also aim to assess the PIVC-P’s usability and potential applicability through the continuous involvement of nurses as end-users.

## 2. Materials and Methods

### 2.1. Design and Procedures

A mixed-design study was conducted in two phases to develop and assess the usability of the PIVC-P, mirroring current recommendations on medical device development and testing [22]. In phase one, during the early stages of concept and semi-functional prototype assessment, two focus groups with nurses were carried out on different dates. Due to the imposition of COVID-19 restrictions at the time in Portugal, these were conducted online by the lead researcher (via Microsoft Teams, and audio recorded via the software-built option).

The first focus group (*n* = 4) revolved around the concept of a PIVC-P and its potential usefulness and applicability in a clinical setting. According to the feedback provided, a semi-functional PIVC-P prototype was developed and assessed by the nurses (*n* = 13) in the second focus group. After a brief period of interaction with the prototype, nurses were asked to fill out a usability questionnaire [23]. Nurses were then asked to discuss the pack’s current requirements, potential use in clinical settings, and design. Any modification suggestions that emerged from this round were discussed, and a note was made for future integration. At the end of phase one, a fully functional prototype was designed that mirrored all the features and specifications discussed and advised by the nurses. 

In phase two, pre-clinical usability assessment tests were conducted with the functioning prototype of the PIVC-P through a two-arm crossover randomised controlled trial (RCT) in a simulation lab. Thirteen nurses were invited and instructed to insert a short peripheral intravenous catheter on an upper arm simulator, three times using traditional material (arm A), and another three times using the PIVC-P (arm B).

Before commencement, nurses were given a few minutes to familiarise themselves with the surroundings and received standardised instructions and materials to perform the usability test. Allocation to each starting arm (A-B or B-A) was pre-determined blindly by a research assistant using the participants’ alphanumerical codes using the online software www.random.org. At the end of phase two, each nurse completed the same usability questionnaire used during the second focus group [23]. Nurses were interviewed to explore their perceptions regarding the usability and applicability of the PIVC-P in a real clinical setting.

### 2.2. Setting and Participants

Recruitment was conducted in April 2020 in a large oncology hospital in Portugal. This institution was primarily selected due to the ongoing implementation of a large action research study within the scope of the PIVC. A participant pool was created with the assistance of a specialist nurse from the hospital, who first contacted nurses and introduced them to the study. 

Participants were eligible to participate if they: (i) held, at least, a nursing degree recognised in Portugal; (ii) were part of the nursing team from the institution’s surgical ward or operating room; (iii) were willing and consented to participate in the study after being presented with its objectives and procedures. The research team randomly selected nurses from the pool, contacted them via e-mail, and explained the details of the project. Where the nurse was willing to participate, further instructions were provided by the research team.

The optimal sample size of usability evaluation is not consensual. Although several authors claim that four or five participants can detect 80% of usability problems, others believe such a sample size may not be sufficient due to the different methods and experience of the test evaluators. Based on predictive data [24], a group of 10 ± 2 participants is recommended to detect 80% of usability problems. Thus, 15 nurses from the initial pool were invited to participate in the study, but due to professional commitments, a final sample of 13 nurses enrolled in the pre-clinical usability assessment tests. 

### 2.3. Materials

In phase two, the pre-clinical usability assessment tests were conducted in the Nursing School of Coimbra’s Simulation Centre. The chosen laboratory simulates two commonly found areas in a typical Portuguese hospital ward: (i) equipped treatment/drug preparation room; (ii) a fully equipped two-bed patient room (Figure 1). 

Both areas contain inbuilt audio-visual recording equipment, which allowed the research team to posteriorly review nurses’ practices during Catheter insertion. PIVC was performed in an upper arm simulator (Multi-Venous IV Training Arm Kit; Nasco Healthcare, Fort Atkinson, WI, USA), that allows for needle insertion and administration of intravenous therapeutics. Potential palpable veins can be located from the antecubital fossa to the dorsum of the hand, with palpable median, basilic, and cephalic veins. 

### 2.4. Outcomes and Measurements 

Usability assessment tests are required to ensure that medical devices fulfil international legal requirements before being introduced in clinical settings. Thus, usability tests should be conducted to assess primary (effectiveness, efficiency, and safety) and secondary outcomes (intention to use and satisfaction). In this pre-clinical study, the measured primary outcomes were: (i) the number of tasks and goals achieved; (ii) procedure execution time; (iii) type and frequency of errors (as per Table 1). The tasks were listed by the research team in accordance with current standards of care in PIVC and recommendations from industrial manufacturers [25,26,27,28]. 

All parameters were assessed through the analysis of the video recordings by the research team and compared to a detailed procedure checklist adapted from a previous study. 

The participants were requested to perform PIVC three times in each arm given the potential existence of a learning curve and training needs required to use the PIVC-P. Secondary parameters were assessed through a usability questionnaire [23], as well as through the audio recordings from the individual interviews and focus group carried out. The questionnaire was comprised of 42 items divided into four domains: (i) usefulness, with 12 items (e.g., item 4: “It helps me to be more efficient”, item 9 “It allows me to have better control over my tasks”); (ii) ease of use, with 10 items (e.g., item 14: “It is simple to use“, item 18 “It does not require physical effort to use”); (iii) ease of learning, with 6 items (e.g., item 25: “I easily remember how to use it”, item 28: “there is no need for written instructions to use it”); and (iv) satisfaction/intention to use, with 14 items (e.g., item 30: “I would recommend it to colleagues”, item 41: “I will like to use it frequently”).

Each item is rated in a 7-point Likert scale (from 1—totally disagree, to 7—totally agree). The original version of the questionnaire evidenced sounding psychometric properties, with an excellent internal consistency (α = 0.976) and good item-total correlation (all above 0.30). In this study, the questionnaire evidence excellent internal consistency (α = 0.964).

### 2.5. Data Analysis

Nurses’ demographic (age, gender), educational (academic degree, specialisation), and professional (clinical experience, work setting, number of PIVC attempts in the last month) data were entered and analysed using the 22.0 version of the Statistical Package for the Social Sciences (SPSS; Chicago, IL, USA). Quantitative data were analysed using descriptive statistics such as means, standard deviations, frequencies, and percentages. Qualitative data gathered during the focus groups and interviews were transcribed using Microsoft Word by two independent researchers, who were unaware of the study goals and participants’ data at the time. Thematic content analysis was performed following the Bardin technique [29]. The emerging categories and subcategories were discussed among the researchers to develop their structure. The recordings and transcriptions were made available to the participants. The findings of this study are reported according to the principles of the GRAMMS guideline [30].

### 2.6. Ethical Consideration

Before the study commencement, the eligible participants were informed by the lead researcher about the study objectives and procedures before they provided written consent. All ethical and legal principles were assured throughout the study, ensuring the participants’ rights to privacy, confidentiality, and the option to leave the study at any time. The audio-visual recordings did not capture any identifiable elements of the participating nurses (e.g., name or face), in order to preserve their anonymity. All data collection instruments were coded alphanumerically per participant and stored in a locked cabinet with restricted access. This study was favourably reviewed by the Ethics Committee of the Health Sciences Research Unit: Nursing of the Nursing School of Coimbra (ref. 617/10-2019).

## 3. Results

### 3.1. Phase 1

During the focus group rounds, nurses were vocal about their opinions on the PIVC-P’s applicability, requirements, and design. Most changes made to the PIVC-P between the first round (concept analysis) and the second round (assessment of a semi-functional PIVC-P prototype) involved the components included in the pack (Table 2).

Concerning PIVC-P requirements, all nurses believed that the prototype should consist of a medium-sized package, sterile sealed, that includes all the required material for safe catheter insertion. According to the participants, all the material inside should also be packaged to avoid clinical waste. Regarding its design, one nurse suggested that the PIVC-P should have a transparent side, to allow healthcare professionals who are not familiar with the device to quickly understand what material is inside. All nurses agreed that the PIVC-P package should have a coloured label according to the gauge of the catheter inside (e.g., if the pack has a 20G PIVC, the label should be pink). Nurses emphasised the need to indicate an expiration date for the pack, as well as a description of the material included. All nurses agreed that the PIVC-P label should include a reminder for health professionals to perform hand hygiene before opening it, as well as to prepare a pair of clean gloves and a sharps container for a safe catheter insertion. 

### 3.2. Phase 2

#### 3.2.1. Laboratory Testing

The participant group (*n* = 13) was mainly composed of female nurses (84.6%), with an average age of 39.5 years (±7.9; 27–59). Participants had an average of 16.7 (±7.3; 5–34) years of nursing experience. A significant number held a master’s degree (46.2%), while others possessed other academic titles such as a graduate degree in nursing (30.8%), a PhD (15.4%), or a bacharelato (7.7%). Eight nurses had a specialisation degree in different areas of nursing science. 

During the performance of PIVC with traditional material (group A), nurses spent an average of 295 (±84.1, 209–503) seconds to execute the required tasks. While using the PIVC-P (group B), nurses spent 246 s (±67.1, 161–376), constituting a statistically significant decrease in procedural time (Z = −2.482, *p* = 0.013). While performing PIVC at the patient bedside, nurses in group A had to stop the procedure 50 times due to the omission of various materials and go back to the treatment room, averaging 3.8 (±2.0; 1–8) lapses per participant. In group B, this number decreased to 17 times, averaging 2.2 (±1.8; 0–5) omissions per nurse, representing a statistically significant decrease (Z = −1.977, *p* = 0.048). In group A, the main areas of divergent practices concerned the non-adherence to the aseptic non-touch technique (ANTT), risk of contact with patients’ blood, the substandard performance of PIVC flushing, and inadequate PIVC fixation. In group B, the main areas of concern were the reuse fabric tourniquet, the non-compliance with the push–pause technique during PIVC flushing, and omission of the catheter insertion date in the dressing. 

Concerning the results of the usability questionnaire, the adjusted global scores (total and domains) of the usability questionnaire filled by the participants after both phases can be found in Table 3. In both phases, the usability scores were consistently higher than the mean ponderation value of the 7-point Likert scale, generally indicating a good usability evaluation of the PIVC-P. Comparing the results obtained with the usability questionnaire between phases I and II, there were statistically significant differences in the mean scores obtained for all dimensions. 

#### 3.2.2. Interviews

Four main categories were identified during qualitative data analysis: (i) time efficiency; (ii) standardization of care delivery; (iii) care safety; and (iv) care sustainability.

Overall, nurses found positive aspects regarding the usefulness and applicability of the PIVC-P in real clinical settings. Several participants identified the PIVC-P as a device that saves them time during the preparation of the required material:


*“Having a pack is extremely functional, especially given the time you save when preparing the material needed for catheterisation […] there is always an item that you forget, and the procedure is not as uniform between professionals as it should be.”*
[Specialist nurse, female, 13 years of professional experience];


*“I really enjoyed the pack. It is practical to have all the material in a single package. It gives us time to concentrate on what really matters to us, which is the person in need of the catheter, not the procedure itself.”*
[Specialist nurse, female, 15 years of professional experience];


*“It is faster! […] Everything in the PIVC-P…we do not have to waste time preparing the flush since the syringe is pre-filled.”*
[Registered nurse, female, 14 years of professional experience];


*“You just need to grab the pack! It saves us time in the selection of the required material […] it also saves us time in something that we often do not do, which is checking the material’s integrity.”*
[Specialist nurse, male, 16 years of professional experience];


*“I do not waste time grabbing material from different cabinets of the treatment room…sometimes from different rooms.”*
[Specialist nurse, female, 16 years of professional experience].

Several nurses highlighted the potential of the PIVC-P in the standardisation of the procedure between nurses, decreasing omissions, as well as enhancing current practices per international standards of care in this field:


*“It allows for the standardisation of the procedure between colleagues (…) we will all start doing the same from now on.”*
[Specialist nurse, female, 13 years of professional experience];


*“It avoids errors and omissions, which often leads to increased procedural times”*
[Specialist nurse, female, 11 years of professional experience];


*“Having a pre-filled syringe (…) people will use it because it is in the pack (…) it reiterates the importance of performing PIVC flush.”*
[Specialist nurse, male, 16 years of professional experience];


*“Most of the required material is collected in a single pack, so I will not be forgetting so many devices.”*
[Specialist nurse, female, 15 years of professional experience];


*“I do not have to constantly worry and think about what I am missing, what is required for catheter insertion.”*
[Specialist nurse, female, 16 years of professional experience];


*“It contains the material that is essential for catheter insertion.”*
[Registered nurse, female, 34 years of professional experience].

Several participants claimed that using the PIVC-P can potentially improve care safety by enhancing current infection control practices:


*“I can only see advantages in its use, especially concerning the antiseptic solution bottle that we use is contaminated or not. With the PIVC-P there is an impregnated swab for each person, and we do not have to manipulate different materials between patients (…) the material would not be circulating between patients.”*
[Specialist nurse, female, 13 years of professional experience];


*“Inside the PIVC-P everything will be sterile and ready to be used (…) it is a way to avoid likely cross-infection that unfortunately happen (…) our tourniquets, as you know, are stored in certain locations that enhance cross-contamination.”*
[Registered nurse, female, 34 years of professional experience];


*“Everything inside the PIVC-P is packaged and sterile, allowing for a great control of potential infection risks.”*
[Specialist nurse, male, 11 years of professional experience];


*“It is safer!”*
[Specialist nurse, female, 11 years of professional experience];


*“More hygienic (…) We do not have to select material from different locations, which often translates into more people manipulating the same materials. Since everything is included in the PIVC-P, we know that it should be disposed of after using.”*
[Registered nurse, female, 14 years of professional experience].

Nurses also praised the contributions of the PIVC-P in the management of existing stock and likely contributions to a sustainable provision of care:


*“Another advantage is the volume of materials that exist in a ward (…) the PIVC-P can facilitate stock management and space consumption.”*
[Specialist nurse, female, 13 years of professional experience];


*“The material inside the PIVC-P (…) everything required is there. We do not have to select materials that, although not used, were opened inside the patient unit, and have to be disposed of.”*
[Registered nurse, female, 34 years of professional experience];


*“The pack was sterile sealed, and all included material has individual packaging (…) I can use everything later.”*
[Specialist nurse, female, 11 years of professional experience];


*“I do not believe that the PIVC-P will make stock management more difficult… everything within the pack can be used later if not open.”*
[Registered nurse, female, 15 years of professional experience];


*“There is no waste! If I open the PIVC-P and do not use a certain material, I can always reuse it later due to the individual packaging.”*
[Specialist nurse, female, 11 years of professional experience];


*“There is less waste.”*
[Specialist nurse, female, 16 years of professional experience].

Finally, we asked the participating nurses if they would change any of the included material in the PIVC’s semi-functional prototype. Although small variations were discussed concerning the potential dismissal of the band-aid or re-addition of the protective field, all nurses converged on the necessity of including a single-use disposable tourniquet:


*“It makes sense because, even if we sanitise current tourniquets, the process is not truly effective.”*
[Specialist nurse, female, 13 years of professional experience];


*“I am unsure in terms of cost but, if possible, the tourniquet should be included within the PIVC-P (…) we just need to grab a pack and it is one less thing to search for.”*
[Specialist nurse, female, 15 years of professional experience];


*“If [single-use disposable tourniquets] are effective, yes, I would include it inside the PIVC-P.”*
[Registered nurse, female, 34 years of professional experience];


*“I believe these tourniquets can avoid contaminations and healthcare-associated infections associated with tourniquet use.”*
[Specialist nurse, male, 11 years of professional experience];


*“If the PIVC-P could include [single-use disposable tourniquets] that would be a great thing”*
[Specialist nurse, female, 11 years of professional experience];


*“At the moment, a disposable tourniquet would add value (…) yes, one inside the PIVC-P.”*
[Registered nurse, female, 14 years of professional experience].

After the final interviews, nurses’ feedback allowed for the refinement of the functional prototype. The final version of the PIVC-P is composed of: 1 short peripheral intravenous catheter sizes 18G, 20G, or 22G; 1 band-aid; 1 pre-filled syringe with 0.9% isotonic sodium chloride (10 mL); 2 sterile gauzes of non-woven fabric; 1 sterile gauze impregnated with an antiseptic solution; 1 needless connector with Luer-lock technology; 1 transparent polyurethane dressing with reinforced borders; and 1 single-use tourniquet. 

## 4. Discussion

In Portugal, as in most international settings, nurses are responsible for peripheral intravenous catheter insertion, maintenance, and supervision [3,4,31,32]. Although it constitutes a rather simple invasive procedure, PIVC complication rates remain considerably high across clinical settings and patient cohorts. Recent studies have focused on the impact of continuous professional training and guideline/bundle implementation in this field, with positive but still unsatisfactory results [19,33,34,35]. 

While we believe that compliance with evidence-informed international standards of care plays a substantial role in achieving better PIVC-related outcomes, the impact of the clinical material used in the insertion and maintenance of peripheral intravenous catheters should not be overlooked. In 2018, Jones outlined a trio of elements that impact PIVC outcomes, including material and technology-related variables. According to the author, although certain technology has improved over time, what makes a difference in PIVC outcomes is the clinician’s grasp of the technology and how to apply it [36]. Historically, while nurses provide a unique end-user-centric, patient-focused approach to medical device development, there is little evidence of their involvement during device development [37]. This continues to be true at present, especially in the fields of vascular access and infusion therapy, with only a few published cases concerning intravenous syringes and infusion pumps [38,39,40,41,42]. 

To the best of our knowledge, this was the first study conducted on the development and assessment of a PIVC-P’s usability. Nurses’ involvement since concept assessment allowed for the structured development of an innovative device for catheter insertion, designed according to the continuous feedback provided by the intended end-users. Throughout phases one and two, several changes were proposed relating to the PIVC-P’s requirements and design specifications, mirroring nurses’ concerns with the device’s applicability, efficiency, and ecological sustainability. 

During phase two, although in a simulated environment, statistically significant positive results were found concerning the PIVC-P’s potential to decrease omissions during material preparation (Z = −1.977, *p* = 0.048) and overall procedural time (Z = −2.482, *p* = 0.013). These results support the potential positive impact of introducing the PIVC-P in clinical settings, given that time constraints are reported by nurses as one of the reasons for the delivery of substandard PIVC-related care and supervision [43,44]. Such potential contributions to the delivery of efficient and safe PIVC-related care were also emphasised by the nurses during the focus groups and interviews. 

According to the participants, one of the main contributions of the designed PIVC-P is the assortment of all the required materials for catheter insertion in a sterile sealed package. This substantially reduces the time spent in collecting the required material, as well as decreasing material omissions, which was found to be statistically significant during the crossover RCT. A previous study found that 13% of all catheter insertions were performed without correct and functional material available [20]. According to the authors, most incidents were related to the unavailability of catheters, tourniquets, skin preparation solutions, and appropriate dressings [20]. More recently, one study conducted in Portugal showed that 65.8% of the time, nurses forgot to prepare certain devices that are essential for catheter insertion, such as appropriate dressings and syringes for catheter flush [45]. Such omissions may hinder PIVC quality and safety, given recent evidence on the importance of catheter-to-vein ratios [46], appropriate dressings for catheter stabilisation [34,47], and skin antisepsis [48,49]. 

The participating nurses also emphasised their enthusiasm with the introduction of materials that are not commonly found in healthcare units in Portugal. The introduction of single-use disposable tourniquets can likely decrease the potential contamination of the puncture [50,51,52]. In most units across the country, nurses use textile or rubber tourniquets during catheter insertion. These devices are known to be difficult to decontaminate between patients and can harbour pathogenic microorganisms [52]. 

Nurses also underlined the importance of including a pre-filled syringe with normal saline for catheter flush. A recent study in Portugal showed that nurses flushed peripheral intravenous catheters 67% of the time after inserting them, though volumes and techniques differed significantly between professionals [14]. According to the study findings, nurses draw the required flushing volume from a 1000 mL bag of normal saline that is used throughout each shift. Manually prepared syringes are subjected to more manipulation during the filling procedure, putting them in closer touch with the healthcare staff and increasing the risk of catheter contamination [53]. Previous studies conducted in Portugal indicate that adherence to PIVC flushing is often overlooked by professionals due to a lack of recent training in vascular access, a lack of material for flushing, or user-unfriendly storerooms [54,55]. Thus, nurses’ inclusion of a pre-filled normal saline syringe in the PIVC-P can likely change current practice and related outcomes, since these devices have shown to be a protective independent factor for PIVC failure [53].

While the PIVC-P seems to assist nurses and other healthcare professionals in the delivery of safe and quality care, its implementation alone will not be sufficient to address the existing high complication rates in Portugal and elsewhere in the world. The PIVC-P should be introduced gradually in a clinical setting, with the understanding that there would be a learning curve for professionals. Other educational initiatives, such as the introduction of clinical bundles and visual reminders of good practices should be encouraged alongside the implementation of the PIVC-P by clinical managers [17,33,56,57]. According to Irvine and colleagues’ Nursing Role Effectiveness Model [6], we believe that the PIVC-P could be an effective organisational resource (structure) that facilitates nurses’ role in PIVC care delivery (process), resulting in improved patient outcomes.

### Limitations

Study findings must be analysed considering its limitations. First, few participants used the “think aloud” strategy during the testing of the PIVC-P in the simulation laboratory, despite being informed of its relevancy by the lead researcher. The “think aloud” strategy enriches data collection by allowing for the research team to understand the struggles faced by users during the use of the device in real-time. However, the final interview conducted right after the laboratory testing was useful in exploring nurses’ perceptions of the PIVC-P’s usability and applicability. Second, although nurses are responsible for PIVC-related care across most of the international settings, future usability studies must be carried out with the involvement of other healthcare professionals who can legally insert peripheral intravenous catheters (e.g., doctors). This will lead to the development of a device that can be used in a standardised way between professionals from different backgrounds. 

Finally, this study was conducted in a simulated environment. Despite the positive results found, future studies in a clinical setting are warranted to assess the PIVC-P’s effectiveness in reducing procedural times and material omission. Future studies must also assess the impact of the PIVC-P in the standardisation of the procedure, as well as its potential impact on the reduction of related complications such as phlebitis or infiltration/extravasation. 

## 5. Conclusions

The involvement of nurses as end-users made it possible, early on, to develop a PIVC-P with perceived usefulness and practical applicability, that enhances the standardisation and systematisation of practices according to the most recent evidence, as well as allows professionals to maximise their time. Future studies in real-world clinical settings are needed to determine the impact of this new device on healthcare professionals’ PIVC practices and outcomes.

## Figures and Tables

**Figure 1 ijerph-19-11130-f001:**
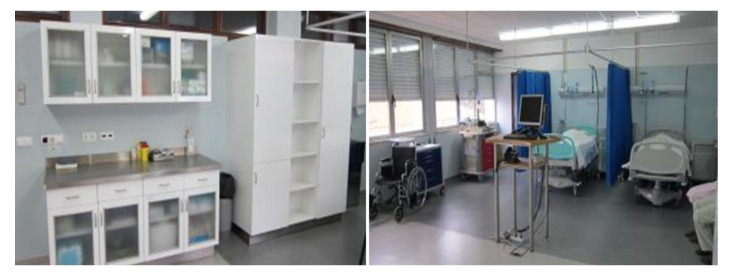
Laboratory setting where the pre-clinical usability assessment tests were conducted.

**Table 1 ijerph-19-11130-t001:** Tasks to be accomplished by the study participants.

Group A (Traditional Material)	Group B (PIVC-P)
Select the following material: 20G catheter; transparent dressing; needless connector; 5 or 10-mL syringe filled with 0.9 isotonic sodium chloride; tourniquet; material for skin antisepsis (gauze and disinfectant); sharps bin container; pair clean gloves; protective field. Take the material to the patient room in a tray. Select an observable/palpable vein. Open the individual packaging of the selected material and dispose of the material as necessary. Place the tourniquet 5–10 cm above the selected puncture site. Perform skin antisepsis. Don clean gloves. Insert the PIVC with a 10–30° angle to the skin. Observe blood return. Apply pressure to the vein (proximally), retrieve the steel needle, and dispose of it in the sharps bin container. Perform catheter flush using the push–pause technique and observe for potential leakage or swelling. Connect needless connector. Remove and dispose of the gloves. Apply transparent (dated), leaving the insertion site visible. Collect the material and go back to the treatment room.	Select the following material: PIVC-P (with 20G catheter); tourniquet; sharps bin container; pair clean gloves; protective field. Take the material to the patient room in a tray. Observe and palpate the venous network to select a suitable vein. Open the PIVC-P and dispose of the included material as necessary. Place the tourniquet 5–10 cm above the selected puncture site. Perform skin antisepsis. Don clean gloves. Insert the PIVC with a 10–30° angle to the skin. Observe blood return. Remove the steel needle and dispose of it in the sharps bin container. Perform catheter flush using the push–pause technique and observe for potential leakage or swelling. Connect needless connector. Remove and dispose of the gloves. Apply transparent dressing (dated), leaving the insertion site visible. Collect the material and go back to the treatment room.

**Table 2 ijerph-19-11130-t002:** Changes to the PIVC-P components between the first and second focus group.

Round 1	Round 2	Reasoning for Change
2 peripheral intravenous catheter sizes 18G, 20G, or 22G	1 peripheral intravenous catheter sizes 18G, 20G, or 22G	First-attempt success rate is high. If difficult catheter insertion is expected, nurses should prepare a second catheter separately.
1 band-aid	1 band-aid	Not applicable.
1 pre-filled syringe with 0.9% isotonic sodium chloride	1 pre-filled syringe with 0.9% isotonic sodium chloride (10 mL)	After discussing current practices and flushing recommendations, nurses believed a 10 mL syringe will not only allow for a post-catheter insertion flush, but also for a PIVC flush if an immediate drug administration occurs.
2 sterile gauzes of non-woven fabric	2 sterile gauzes of non-woven fabric	Not applicable.
1 sterile gauze impregnated with antiseptic solution	1 sterile gauze impregnated with antiseptic solution	Not applicable.
1 pair of clean gloves	Not applicable.	Glove size depends on user, creating the need for a larger number of PIVC-P (per catheter and glove size). Gloves should be prepared separately.
1 needless connector	1 needless connector with Luer-lock technology	Safer connection with syringes, less risk of catheter accidental removal or dislodgement.
1 transparent dressing	1 transparent polyurethane dressing with reinforced borders	Better fixation, allowing for patients to carry their basic activities of daily living (e.g., showering) without risking accidental PIVC removal.
1 protective field	Not applicable.	Not commonly used, generates clinical waste. Blood spillage is reduced given the technology of the selected PIVC.

**Table 3 ijerph-19-11130-t003:** Differences in usability scores between phase 1 and phase 2.

Dimensions	Semi-Functional Prototype Assessment (SD; min. − max.)	Functional Prototype Assessment (SD; min. − max.)	Differences
Usefulness	M = 6.25 (±0.67; 5–7)	M = 6.75 (±0.38; 5.67–7)	Z = −2.492 (*p* = 0.013)
Ease of use	M = 6.13 (±0.59; 5–7)	M = 6.81 (±0.32; 5.9–7)	Z = −2.805 (*p* = 0.005)
Ease of learning	M = 5.87 (±0.97; 4–7)	M = 6.74 (±0.39; 5.67–7)	Z = −2.527 (*p* = 0.012)
Satisfaction/intention to use	M = 6.32 (±0.57; 5–7)	M = 6.87 (±0.17; 6.57–7)	Z = −2.937 (*p* = 0.003)
Total score	M = 6.19 (±0.59; 5–6.9)	M = 6.80 (±0.22; 6.36–7)	Z = −3.041 (*p* = 0.002)

Note. M = mean; SD = standard deviation; min. = minimum value; max. = maximum value; Z = Wilcoxon signed-rank test.

## Data Availability

The raw data supporting the conclusions of this article will be made available by the authors upon request, without undue reservation.

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
