# Peer review of "Nurses’ Involvement in the Development and Usability Assessment of an Innovative Peripheral Intravenous Catheterisation Pack: A Mix-Method Study"

_ijerph, 2022, doi:10.3390/ijerph191711130_

Round 1
Reviewer 1 Report
The goal of the project was: „to develop a Peripheral intravenous catheterization pack (PIVC- P) that follows current international standards of care in vascular access in adult patients and can be implemented in clinical settings across Portugal and to assess the PIVC-P's usability and potential applicability through the continuous involvement of nurses as end-users”.
It is an interesting and innovative research project with great potential for impacting safety practices and reducing complications (such as phlebitis or infiltrates, for example) in using Peripheral intravenous catheterization (PIVC)in healthcare with great commitment in practice by nurses.
1. Study participants recruitment description (point 2.2 Setting and participants): the description provided does not meet the requirements of the indicated methodology of a two-arm randomized controlled trial (RCT) for researchin phase two pre-clinical (point 2.1. Design and procedures). In randomized controlled trials, trial participants are randomly assigned to either treatment or control arms. Please describe in detail the procedure and method of selecting nurses for the study (what was the database of nurses, method for random selection, etc.)
2. Please provide a more precise description of the usability questionnaire [42 items divided into four domains: usefulness, ease of use, ease of learning, and satisfaction/intention to use]: please give the number of items in each domain, and give sample questions.
3. The qualitative analysis presents interesting results, although difficult to take in. It would be worthwhile at the beginning to list the topics extracted from the interviews and then describe them.
4. There is also no attempt to integrate the data and conclusions from various components within the quantitative and qualitative research, which is the essence ofa mixed-method study.
Author Response
We appreciate Reviewer 1's time dedicated to the peer-review of the manuscript. Below are the observations made by the reviewer and our replies.
Observation 1: Study participants recruitment description (point 2.2 Setting and participants): the description provided does not meet the requirements of the indicated methodology of a two-arm randomized controlled trial (RCT) for researchin phase two pre-clinical (point 2.1. Design and procedures). In randomized controlled trials, trial participants are randomly assigned to either treatment or control arms.
Reply 1: We appreciate this comment, which may have been caused by the lack of clarity on our part. The controlled trial followed a crossover design, often used during the development of medical devices to compare how the same individuals behave/operate with different conditions. This means that the same sample will experience both the control and experimental conditions, although the order must be random (e.g., A-B or B-A; A-B-C, B-C-A, B-A-C, A-C-B).
According to international recommendations, a sample of 8 to 12 elements is required during this phase to account for 80% of all usability issues (as stated in our manuscript). Our study sample (n = 13) was achieved after contacting a pool of 15 interested nurses, who work in the hospital where this project was being conducted, as also stated. Due to scheduling conflicts, 13/15 participated in the study.
Nonetheless, to improve the manuscript's clarity, we have included this information in the method section (highlighted in track change).
Observation 2: Please provide a more precise description of the usability questionnaire [42 items divided into four domains: usefulness, ease of use, ease of learning, and satisfaction/intention to use]: please give the number of items in each domain, and give sample questions.
Reply 2: We appreciate this request and have included the changes on page 5, lines 174-181 (highlighted in track change).
Observation 3: The qualitative analysis presents interesting results, although difficult to take in. It would be worthwhile at the beginning to list the topics extracted from the interviews and then describe them.
Reply 3: We agree with Reviewer 1 and have included the changes on page 8, lines 263-265 (highlighted in track change).
Observation 4: There is also no attempt to integrate the data and conclusions from various components within the quantitative and qualitative research, which is the essence of a mixed-method study.
Reply 4: We appreciate Reviewer 1's comment.
Given the length of the manuscript and the amount of data, we tried to keep the discussion straightforward, which was appreciated by the other reviewer. However, outcomes were both quantitative and qualitative data were collected (e.g., the effect of the designed PIVC-P on procedural times and material omissions) were discussed in an integrative way.
For example, discussing some of the qualitative feedback (e.g., interviews) in light of some of the quantitative data (e.g., questionnaire - satisfaction/intention to use dimension) would increase the number of pages substantially and without adding to what is already explicit (in our opinion). Although we

Reviewer 2 Report
This is an excellent paper describing focus groups and interviews with nurses to develop and assess the usability of an innovative PIVC pack, a device with a significant clinical need for improvement, internationally, as stated clearly by the authors. There is a thorough and logical review of the relevant and recent literature in a well structured introduction. Appropriate methodology is employed and explanations given for study design. Throughout the paper, the current body of literature is appropriately cited in a comprehensive manner. Methods and results require minor clarifications for ease of understanding for the reader. Conclusions drawn are appropriate to the results found, which clearly show that involving nurses resulted in significant improvements in ease of use with the prototype developed based on their initial suggestions and comments. This impactful work adds significant value to the field, which is lacking in innovations and usability assessments in infusion therapies and devices.

Author Response
We deeply appreciate the reviewer's time and constructive feedback, as well as assistance with semantics and grammar. This was one of the best peer-review experiences we have had in a long time - thank you!
Most recommendations were included as per the reviewer's feedback (all highlighted with track changes in the manuscript). Given the long list provided by Reviewer 2, we will only address below the suggestions/comments that were not integrated/required:
Observation 1: Line 133: Consider adding reference to whether this likely mirrors European/international settings also.
Reply 1: We understand the request, but could not find a reference that supports such likeliness. Therefore, we did not change the sentence.
Observation 2: Line 175: Ensure you have the appropriate reference for GRAMMS guideline
Reply 2: We believe the reference is correct since we assessed the GRAMMS recommendations directly through the EQUATOR network (https://www.equator-network.org/reporting-guidelines/the-quality-of-mixed-methods-studies-in-health-services-research/). The provided reference matches ours.
